# Evaluation of Different Potassium Management Options under Prevailing Dry and Wet Seasons in Puddled, Transplanted Rice

Suchismita Mohapatra [1,2], Kumbha Karna Rout [2], Chandramani Khanda [2], Amit Mishra [3], Sudhir Yadav [4], Rajeev Padbhushan [5], Ajay Kumar Mishra [6,*] and Sheetal Sharma [7,*]

1 Siksha' O'Anusandhan, Deemed to Be University, Bhubaneswar 751003, India
2 Orissa University of Agriculture & Technology, Bhubaneswar 751003, India
3 Banda University of Agriculture & Technology, Banda 210001, India
4 International Rice Research Institute, Metro Manila 7777, Philippines
5 Bihar Agricultural University, Sabour 813210, India
6 International Rice Research Institute South Asia Regional Centre, Varanasi 221106, India
7 International Rice Research Institute, New Delhi 110012, India
* Correspondence: a.k.mishra@irri.org (A.K.M.); sheetal.sharma@irri.org (S.S.)

**Abstract:** The present field experiment was conducted in both dry season (DS) and wet season (WS) from 2014–2015 to evaluate the influence of different potassium (K) management options (graded doses of inorganic K fertilizer alone and combined with foliar and straw incorporation) on the rice yield, nutrient uptake, and soil K balance under puddled, transplanted rice in acidic soil. The results showed that rice yields were higher under WS as compared to the DS crop. Among treatments, $K_{40} + K_{spray}$, i.e., the combination of inorganic K fertilizer (40 kg $K_2O$ ha$^{-1}$) along with a foliar spray of K (1% $KNO_3$) at the panicle initiation stage, produced the highest grain yield in both seasons; however, it was on par with treatments $K_{80}$, i.e., the highest dose of inorganic K fertilizer (80 kg $K_2O$ ha$^{-1}$) alone, and $K_{30} + K_{straw}$ i.e., integrated use of inorganic K fertilizer (30 kg $K_2O$ ha$^{-1}$) and straw (3 t ha$^{-1}$, 45 kg $K_2O$ ha$^{-1}$). Application of 80 kg $K_2O$ ha$^{-1}$ through inorganic fertilizer alone had the maximum K uptake at the harvest stage in both seasons. DS rice had a higher K/N and K/P ratio than the WS. The treatments applied with inorganic K fertilizers, either soil or foliar applications, had negative K balance in both seasons; however, treatments applied with organic sources of K, i.e., rice straw alone or integrated with inorganic K fertilizers, had positive K balances in the soil. Therefore, this study shows that the integrated use of inorganic K fertilizer and 3 t ha$^{-1}$ rice straw ($K_{30} + K_{straw}$) can be a recommended option for a better K management strategy for crop yields and soil sustainability in acid soils. However, in terms of greenhouse gas (GHG) estimation, incremental doses of soil-applied K fertilizer along with straw aggravate the GHGs emission in the rice–rice cropping system, and among all treatments, $K_{40} + K_{spray}$ is the promising treatment which requires intensive investigation for drawing an overall conclusion.

**Keywords:** soil-applied potassium; foliar spray; rice straw; potassium uptake; potassium balance; potassium use efficiency

## 1. Introduction

Rice (*Oryza sativa*) is the staple food for more than half of the world's population. Globally, rice is the second most crucial cereal crop cultivated in a 164-million-ha area, with a worldwide production of 757 million tons per ha and an average productivity of 4609 kg per ha [1,2]. It is pivotal in providing food and livelihood security to rural people, particularly in Asia. In India, rice occupies 35% of the total food grain area and contributes 40% of the food grain production. Odisha, a state in eastern coastal India, occupies only 3% of India's total rice area while sharing 7% of India's rice production. As per the estimates of population growth, there will be a deficit of about 2.5 million tons of rice if the present supply growth rate continues till 2030 [3]. Rice production faces certain challenges such

as low and imbalanced use of fertilizers, low fertilizer use efficiency, insect and disease infestation, late arrival of monsoons, and natural disasters such as floods, drought, etc., affecting production and productivity at a large scale [4]. Therefore, rice production needs to be increased sustainably by applying proper management practices without significantly impacting the environment.

The climatic conditions of eastern and southern coastal India are favorable for rice production in both DS and WS. DS starts in November and ends in March; however, WS begins in June and finishes in October. WS receives around 90–91% of the total annual rainfall, while only 5–6% is received during DS. Lower temperatures and humidity (around 70%) during the DS crop period coincide with the early vegetative growth stage, resulting in the poor availability of inherent nutrients, in turn impacting the agricultural production system. Due to the high groundwater table and assured irrigation during the DS, farmers in the region (eastern and southern coastal India) also prefer to grow rice in DS along with WS. Although, intensive cropping systems and the use of high-yielding varieties have increased the rice yields but disturbed the soil nutrient status [5–7] because of the imbalanced use of fertilizers. Therefore, a proper nutrient management strategy is essential for a sustainable rice production system.

Potassium is often the most limiting nutrient after nitrogen (N) in high-yielding fertilizer-responsive rice systems; however, it is paid less attention due to its slow effect in increasing yields and non-polluting nature [8,9]. Crops are more susceptible to many plant diseases when K nutrition is low, which can cause harvest and quality losses beyond the physiological effects of inadequate K nutrition. In contrast to N and phosphorus (P), K fertilizers are applied at a much lower rate which is less than 50% of the total K removed by crops [10]. In the absence of an adequate external supply, plants will fulfill their requirement from the native reserve K present in the soil, leading to issues such as depletion of soil K reserves, which is mainly undetected by the conventional soil tests for K and very poor responses to the small amount of K applied through fertilizers. Continuous K mining increases the K fixation capacity of soils, particularly of coarse-textured acid Alfisols [11] found in eastern India and used mainly for a continuous crop of rice, i.e., DS and WS, which is considered the major reason for non-responses or feeble responses to the small amount of K applied.

In India, there was no or negligible response to K application in rice during the 70s [12,13] due to the high K-supplying capacity of the soil, whereas, after the 70s, introducing a high-yielding fertilizer-responsive system depleted the K soil resources, and crops' response to K increased extensively [14]. Large areas of the world's arable soils are deficient in K due to the low application rate of K fertilizer [15].

Studies on various soils of eastern Indian states have reported low or deficient K levels [16–18]. In Odisha, red and yellow acidic soils are dominant and inherently poor in nutrients, particularly in K [19]. This situation has been exacerbated due to the increased use of N fertilizers for better yields [20]. Now, some nutrient-exhaustive crops have started exhibiting symptoms of K deficiency. However, the K-supplying capacity of soil under the rice–rice system needs to be understood for a judicious recommendation of K fertilizer for better economic returns and ensuring soil sustainability through maintaining soil fertility [21].

The sustainability of crop production is not possible unless we check K depletion, which is possible by adding a matching amount of K either by increasing the recommended dose of K fertilizer (40 kg ha$^{-1}$) or through other sources such as paddy straw, which contains a large amount of K (1.5%) and other nutrients and is readily available as compared to organic manures such as FYM. With the poor response of crops, the nutrient adequacy can be tested by the supply of K through foliar feeding and studying the nutrient ratio involving K (K/N, K/P). The K reserves of India are penurious, and a massive quantity of K fertilizers is imported from other countries, which ultimately increases its market cost. Thus, alternative K sources need to be investigated for their K utilization efficiency, which could be supplemented along with K fertilizer.

Rice stubbles or straws are loaded with K, and full utilization of the straws can solve the problem of the K crisis. Straw incorporation in the soil to improve the soil's physical and chemical properties, in addition to returning a significant amount of K into the soil, were reported by several researchers [9,22,23], even though it is not popular in rainfed rice cultivation regions. Moreover, the removal of K has been exacerbated by the practice of rainfed-area resource-poor farmers, who remove K from their fields as straw for cattle feed [24–26]. Therefore, relieving the soil K deficit just through soil–plant internal circulation is insufficient. Combining crop straw returns with K fertilizer is best to increase K cycling and balance the K deficit in the soil.

A foliar spray of K salts has been promoted to supply additional K during critical stages of the plants' lifecycle, resulting in increased yield and nutrient uptake in rice [27]. Rice absorbs K until the maturity stage. Hence, foliar application of K in divided doses at critical crop growth stages is essential for increasing yields and nutrient uptake, although it helps the rice plants resist pests and disease [28,29]. Two-three foliar feeding of spring and summer rice crops with $KNO_3$ has provided better yields and net income responses [30].

Hence, evaluating alternative K management practices such as straw incorporation and targeting critical stages through foliar spray along with chemical fertilizers under puddled, transplanted rice–rice systems is essential to maintain soil K balances and yields. So far, limited studies have been performed mainly in acidic soils of eastern coastal India to evaluate K management strategies under both DS and WS in puddled, transplanted rice. This study hypothesized that rice straw incorporation in soil and foliar application of $KNO_3$ either fully or partly meets the K demand of the crop and maintains the K balance in the soil system. Therefore, treatments with higher doses of K (40, 60, 80 kg ha$^{-1}$) and inclusion of straw incorporation and foliar feeding as a supplemental source (not as a substitute) to fertilizer K at the rate of 40 kg $K_2O$ ha$^{-1}$ were considered. These treatments are evaluated in terms of yield response and nutrient balance for managing K nutrition in a highly intensive rice–rice system, a major cropping system in eastern India. K also plays an important role in regulating the production and emission of methane ($CH_4$) and nitrous oxide ($N_2O$) through stoichiometric relations with carbon (C) and N. Previous studies reported that the application of K reduced $CH_4$ emissions from flooded rice fields by mitigating methanogenic bacteria and stimulating methanotrophic bacterial populations [31].

The objectives of this study were to assess the effects of different K management options on the rice yields and yield-attributing characters under both DS and WS in acidic soil, to determine the impacts of K management options on nutrient uptake, K uses efficiencies, and K balance in rice soil, and the estimation of greenhouse gases (GHGs) emission from K management options in rice–rice cropping systems.

## 2. Materials and Methods

### 2.1. Field Site

The present study was carried out at the Central Research Station of Orissa University of Agriculture and Technology (OUAT), Bhubaneswar, India (20°15′ N, 81°52′ E, 25.5 m from msl) during DS and WS. The site comes under the east and south-eastern coastal plains of Odisha. The soil of the experimental site is sandy loam with an acidic pH (5.87), low in organic C (3.9 g kg$^{-1}$) and available N (170 kg ha$^{-1}$), medium in available phosphorus (P) (21 kg ha$^{-1}$), and low in available K (57.95 kg ha$^{-1}$) in the surface layer (0–15 cm). Based on soil analysis, lower layers contained more clay with reduced acidity. Taxonomically, the soil belongs to Inceptisols, is grouped as Vertic Haplaquept, and is mixed hyperthermically. The land remains ill-drained during the wet season because of the shallow water table (1 m), whereas it is moderately well-drained during the DS.

### 2.2. Climate

The total rainfall received during the DS and WS cropping system was 1472 mm, higher than the five-year average (2009–2013), i.e., 1389 mm (Figure 1). WS rice (June–October)

received 91% of total rainfall. However, only 6% of rain was received during the DS crop period. The mean maximum temperature was highest in April–May and reached 40 °C, whereas the mean minimum temperature was low (20 °C) in December–January. There was a low temperature during December and January, which coincided with the early vegetative period of crop growth of DS rice. During the DS, relative humidity was almost near 70%. However, it increased gradually in the WS.

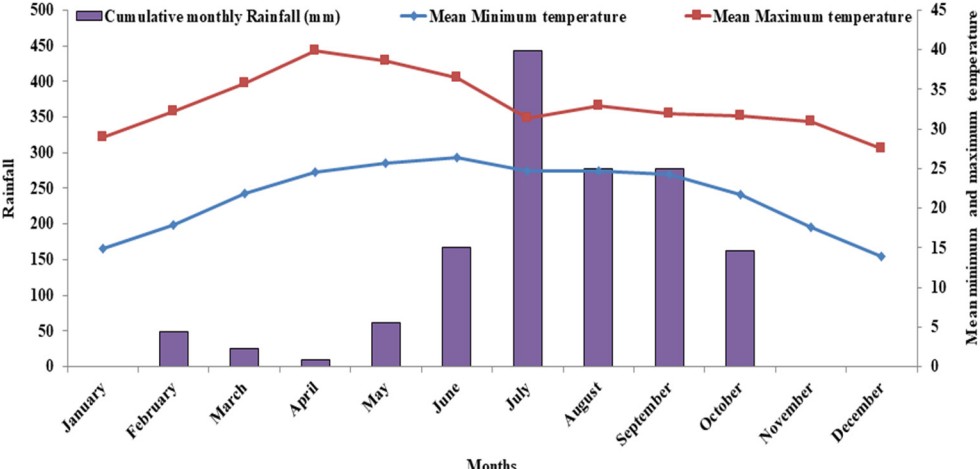

**Figure 1.** Monthly mean minimum, maximum temperature (°C), relative humidity (%), and cumulative monthly rainfall (mm) of the experimental site.

### 2.3. Experimental Design

The experiment was laid out in randomized block design (RBD) with nine treatments: $K_0$ (Without K, control), $K_{40}$ (40 kg $K_2O$ ha$^{-1}$), $K_{60}$ (60 kg $K_2O$ ha$^{-1}$), $K_{80}$ (80 kg $K_2O$ ha$^{-1}$), $K_{straw}$ (45 Kg K through straw ha$^{-1}$), $K_{20}$ + $K_{straw}$ (20 kg $K_2O$ ha$^{-1}$ + 45 Kg K through straw ha$^{-1}$), $K_{30}$ + $K_{straw}$ (30 kg $K_2O$ ha$^{-1}$ + 45 Kg K through straw ha$^{-1}$), $K_{40}$ + $K_{straw}$ (40 kg $K_2O$ ha$^{-1}$ + 45 Kg K through straw ha$^{-1}$), and $K_{40}$ + $K_{spray}$ (40 kg $K_2O$ ha$^{-1}$ + 1% K as $KNO_3$ through foliar spray) replicated thrice. The details of the treatments iare represented in Table 1. The plot size was 30 m$^2$ (6 m × 5 m). For rice, the recommended doses of N, $P_2O_5$, and $K_2O$ are 117, 40, and 40 Kg ha$^{-1}$, respectively.

**Table 1.** Amount of nutrients added in different potassium management practices in this study.

| Treatments | Nitrogen (kg ha$^{-1}$) | Phosphorus (kg ha$^{-1}$) | Potassium (Chemical Fertilizer) (kg ha$^{-1}$) | Potassium (Straw) (kg ha$^{-1}$) | Potassium Nitrate (Foliar) (%) | Zinc (kg ha$^{-1}$) |
|---|---|---|---|---|---|---|
| $K_0$ | 117 | 40 | - | - | - | 5 |
| $K_{40}$ | 117 | 40 | 40 | - | - | 5 |
| $K_{60}$ | 117 | 40 | 60 | - | - | 5 |
| $K_{80}$ | 117 | 40 | 80 | - | - | 5 |
| $K_{straw}$ | 117 | 40 | - | 45 | - | 5 |
| $K_{20}$ + $K_{straw}$ | 117 | 40 | 20 | 45 | - | 5 |
| $K_{30}$ + $K_{straw}$ | 117 | 40 | 30 | 45 | - | 5 |
| $K_{40}$ + $K_{straw}$ | 117 | 40 | 40 | 45 | - | 5 |
| $K_{40}$ + $K_{spray}$ | 117 | 40 | 40 | - | 1 | 5 |

### 2.4. Crop and Nutrient Management

The field experiment was conducted on a typical rice–rice cropping system taking a medium-land- and medium-duration-(120–125 days)-variety 'Lalat'. A power tiller was used to puddle one day of transplanting. The seedling's age was 25 days in DS and 22 days in WS, and it was transplanted in $20 \times 10$ cm spacing. At the time of transplanting, 30 kg N ha$^{-1}$ as urea, 40 kg $P_2O_5$ ha$^{-1}$ as diammonium phosphate (DAP), and 25 kg zinc (Zn) ha$^{-1}$ as zinc sulphate heptahydrate ($ZnSO_4 \cdot 7H_2O$) were applied in all the plots. Another 80 kg N was used in two splits at a ratio of 2:1 at 20 days and 50 days after transplanting (DAT). K in the form of muriate of potash (MOP) was applied as per the treatment in two split doses, 50% at basal and 50% at 50 DAT. A foliar spray of 1% $KNO_3$, was performed at the P.I. stage.

In treatments where straw incorporation was a component of K management, harvesting was completed by removing the panicle attached to a 5 cm straw. The field was then allowed to dry, and the straw was incorporated to a shallow depth using a power tiller. The second ploughing was completed after 15 days to encourage decomposition. Other plots were also ploughed simultaneously. Approximately 3 t ha$^{-1}$ of straw was added.

Pretilachlor @ 500 mL a.i. per ha was applied at 2 DAT, and manual weeding and bund cleaning in later stages were performed for managing weeds in rice fields. The pests and diseases were managed by application of Furadan @ 10 kg per acre, Streptocycline @ 1.5 g per 10 L, and SAAF @ 2 g per liter as a prophylactic measure and again during the peak period damage.

### 2.5. Growth, Yields, and Yield Component Analysis

The plants' biomass was collected at mid-tillering (M.T.), P.I., and maturity stages by cutting twelve hills in two adjacent rows at two spots. The grain and straw yields were calculated based on yields obtained in a 4 m$^2$ area chosen at the center of the field and expressed as t ha$^{-1}$. The grain was harvested and threshed manually. The moisture content of the grain was measured by a moisture meter, and the grain yield was adjusted to 14% moisture content. The moisture content of the straw was determined by taking a small sample of straw of a known weight and oven drying it at 70 °C for 72 h, and the straw yield was adjusted to 0% moisture content. The harvest index was determined by taking the ratio of the grain yields to the biological yields. The yield components measured at physiological maturity included effective tillers per square meter, number of grains per panicle, and 1000-grain weight out of 10 randomly selected plants. Filled and unfilled grains from each sample of 10 randomly selected plants were separated, and the total numbers of filled and unfilled grains were calculated. The percentage of grain filling was determined by dividing the number of filled grains by the total number of grains (filled and unfilled) multiplied by 100.

### 2.6. Plant Sampling and Analysis

The sampling of plants was performed at the M.T., P.I., and physiological maturity stages. From each treatment subplot, 24 hills were sampled, with 12 hills from 2 opposite sites, each consisting of 2 adjacent rows with 6 hills in each row. The hills were cut at 2 cm above ground level. The collected clumps were dried under the sun and kept in the oven at 70 °C $\pm$ 5 °C. Dried plant materials were weighed to determine biomass, chopped into small pieces with stainless-steel scissors, and ground by a Wiley mill for conducting total nutrient (N, P, and K) analysis. The plant samples were analyzed for K content for all the stages. The grain and straw samples were separated at 90 DAT, processed and analyzed for N, P, and K. Plant samples were digested using $HNO_3$-$HClO_4$ (3:2) to determine K by diluting and P by the Vanadomolybdate yellow color method through a flame photometer and spectrophotometer, respectively [32]. N was determined by the micro-Kjeldahl method [32].

The different agronomic parameters [33,34] were calculated using the following equations:

1. Nutrient (N/P/K) uptake by grain/straw $\left(\text{Kg ha}^{-1}\right)$

$$\text{Nutrient uptake by plant} = \frac{\text{Grain/Straw weight}\left(\text{Kg ha}^{-1}\right) \times \text{N/P/K content in (Grain/straw \%)}}{100} \tag{1}$$

2. Harvest index of potassium (KHI)

$$\text{KHI} = \frac{\text{KG}}{(\text{KG} + \text{KS})} \tag{2}$$

3. Agronomic efficiency of K (AEK) (kg kg$^{-1}$)

$$\text{AEK} = \frac{(\text{GYF} - \text{GY0})}{\text{KF}} \tag{3}$$

4. Recovery efficiency of K (REK) (%)

$$\text{REK} = \frac{(\text{TKF} - \text{TK0})}{\text{KF}} \tag{4}$$

5. Ratio between N, P, and K

$$\text{K} : \text{N} = \frac{\text{TK}}{\text{TN}} \tag{5}$$

$$\text{K} : \text{P} = \frac{\text{TK}}{\text{TP}} \tag{6}$$

where,

KG is K uptake by grain, and KS is K uptake by straw;
GY0 is grain yield (14% moisture content) without K application (K0);
GYF is grain yield (14% moisture content) with fertilizer K application (K.F.);
TK0 is the total plant K uptake without K application (Soil K uptake);
TKF is total plant K uptake with fertilizer K application;
TK is the total (grain + straw) K uptake;
TN is the total (grain + straw) N uptake;
TP is the total (grain + straw) P uptake.

### 2.7. Potassium Balance

The K balance was determined from the difference between the total amount of K added through fertilizer, FYM, irrigation, and straw and the removal of K from the aboveground plant part (straw + grain).

### 2.8. Soil Available K

Composite soil samples were collected mid-season, i.e., at M.T. (30 DAT), P.I. (60 DAT), and harvested at depths of 0–15 cm and 15–30 cm. The collected soil samples were then air-dried under shade, crushed, sieved through a 2 mm sieve, and analyzed for 1N NH$_4$OAc K or exchangeable K [35].

### 2.9. Global Warming Potential (GWP) Analysis

Net GWP of rice was estimated by using all the sources and sinks of GHGs such as emissions due to the production and transportation of fertilizers, field operations (tillage, seeding, irrigation), retention/incorporation of crop residues, land use management, soil properties, C-sequestration, and soil flux of GHGs. The emissions of GHGs were computed by using the CCAFS Mitigation Options Tool [36]. In this tool, many empirical models are combined at a regional scale to compute GHG emissions in any production system. The tool considers specific factors, i.e., climatic conditions, soil characteristics, crop production inputs, and other management activities that influence emissions. The background and

fertilizer-induced emissions are estimated using the multivariate empirical model (MEM) of [37] for $N_2O$ and nitric oxide (NO) emissions, and the FAO (2001) model for ammonia ($NH_3$) emissions. Emissions led by crop residues were computed through IPCC $N_2O$ Tier-1 emission factors. Alterations in SOC due to tillage operations, organic inputs, and residue retention/incorporation are based on the IPCC methodology described by [38,39]. The $CO_2$ emissions from soil resulting from urea or liming were calculated as projected by IPCC methodology. GWP of rice–rice systems under different treatments with K fertilizer management was computed on a base GWP (over 100 years) of 298 for $N_2O$ and 34 for $CH_4$ (IPCC 2013).

Global warming potential (GWP) and total GWP were calculated using the equations below.

$$\text{GWP (kg CO}_2 \text{ eq ha}^{-1}) = CO_2 \text{ (kg ha}^{-1}) + N_2O \text{ (kg ha}^{-1}) \times 298 + CH_4 \text{ (kg ha}^{-1}) \times 34 \tag{7}$$

$$\text{Total GWP} = \text{soil C GWP} + \Delta \text{ soil CH}_4 \text{ emission} + \text{soil N}_2\text{O emission} + \text{operation GHG emission} + \text{input GHG emission} \tag{8}$$

### 2.10. Statistical Analysis

The data for different parameters were analyzed using variance (ANOVA) analysis following other statistical procedures [40,41]. The treatment means for two years of data were compared by least significant difference (LSD) at a 0.05 level of probability and represented using the Duncan Multiple Range tests (DRMT) shown in the tables.

## 3. Results

### 3.1. Plant Biomass as Affected by K Management at M.T., P.I., and Harvest Stages

Plant biomass increased with the advancement of crop growth (Table 2). Plant biomass was consistently highest in $K_{40} + K_{spray}$, during the DS. The biomass produced in the WS varied significantly among the treatments at M.T. and the harvesting stages. $K_{80}$ accumulated a substantially higher amount of biomass than most treatments throughout the WS.

**Table 2.** Effect of potassium management practices on biomass accumulation (kg ha$^{-1}$) at various stages of rice crop growth in dry and wet seasons.

| Treatments | Dry Season | | | Wet Season | | |
|---|---|---|---|---|---|---|
| | Mid Tillering | Panicle Initiation | Harvest | Mid Tillering | Panicle Initiation | Harvest |
| $K_0$ | 1450 [ab] | 3970 [ab] | 7100 [b] | 1540 [ab] | 3890 [a] | 8560 [d] |
| $K_{40}$ | 1550 [ab] | 4170 [ab] | 7580 [b] | 1570 [ab] | 3820 [a] | 9190 [cd] |
| $K_{60}$ | 1530 [ab] | 4560 [a] | 8640 [ab] | 1580 [ab] | 4340 [a] | 11,070 [abc] |
| $K_{80}$ | 1510 [ab] | 4250 [ab] | 9780 [a] | 1780 [a] | 4180 [a] | 11,560 [a] |
| $K_{straw}$ | 1210 [b] | 3720 [b] | 7720 [b] | 1350 [b] | 4330 [a] | 9540 [bcd] |
| $K_{20} + K_{straw}$ | 1250 [b] | 3710 [b] | 8130 [ab] | 1870 [a] | 3930 [a] | 8650 [d] |
| $K_{30} + K_{straw}$ | 1350 [b] | 3720 [b] | 8970 [ab] | 1420 [ab] | 4270 [a] | 11,060 [abc] |
| $K_{40} + K_{straw}$ | 1400 [ab] | 3940 [ab] | 8280 [ab] | 1320 [b] | 4330 [a] | 10,300 [bcd] |
| $K_{40} + K_{spray}$ | 1650 [a] | 4580 [a] | 10,060 [a] | 1440 [ab] | 4480 [a] | 11,230 [ab] |

Values denoted with the same letter are not significantly different at $p < 0.05$ using Duncan's Multiple Range Test.

### 3.2. Yield-Attributing Characters

Yield-attributing characters such as the number of effective tillers per square meter, spikelet per panicle, grain filling percentage, and 1000-grain weight are presented in Table 3 for DS and WS There was a significant difference among the treatments concerning yield-attributing characters in the DS, while no significant difference was found in the WS except for the filled grains (%). There was a consistent trend in the DS with better yield-attributing characters in the $K_{40} + K_{spray}$ treatment. Whereas the number of spikelets per panicle was

higher in $K_{80}$, the yield-contributing characters were significantly lower in $K_{straw}$ and $K_0$ in DS. Similarly, 1000-grain weight was significantly higher in $K_{40} + K_{spray}$ than in $K_{straw}$ and $K_0$. The number of effective tillers per square meter was much lower in DS than in WS. In DS the number varied between 227 and 313 compared to 375 and 460 in WS. There were no remarkable differences in yield-attributing characters except the percent of filled grain during the WS. The fertility percentage in WS varied from 82.0% to 87.5%, with a significant difference among the treatments (Table 3).

**Table 3.** Effect of potassium management practices on yields and yield-attributing parameters at various crop growth stages in the dry and wet seasons.

| Treatment | Effective Tillers $m^{-2}$ | No. of Spikelet $Panicle^{-1}$ | Filled Grains (%) | Test Weight (g) | Gr. Yield (t ha$^{-1}$) | St. Yield (t ha$^{-1}$) | Harvest Index |
|---|---|---|---|---|---|---|---|
| | | | Dry Season | | | | |
| $K_0$ | 227 [c] | 87 [b] | 86.1 [bc] | 25.2 [b] | 2.69 [b] | 4.41 [c] | 0.37 [abc] |
| $K_{40}$ | 277 [ab] | 93 [ab] | 86.1 [bc] | 25.3 [b] | 3.21 [b] | 4.38 [c] | 0.42 [a] |
| $K_{60}$ | 294 [a] | 98 [ab] | 90.7 [ab] | 25.7 [ab] | 3.51 [ab] | 5.13 [abc] | 0.41 [ab] |
| $K_{80}$ | 273 [abc] | 101 [a] | 89.6 [abc] | 26.2 [ab] | 3.43 [ab] | 6.35 [a] | 0.35 [c] |
| $K_{straw}$ | 228 [c] | 88 [ab] | 84.6 [c] | 25.2 [b] | 2.82 [b] | 4.90 [bc] | 0.37 [abc] |
| $K_{20} + K_{straw}$ | 238 [bc] | 95 [ab] | 89.3 [abc] | 25.3 [b] | 3.17 [b] | 4.96 [bc] | 0.39 [abc] |
| $K_{30} + K_{straw}$ | 238 [bc] | 95 [ab] | 91.2 [ab] | 25.5 [ab] | 3.48 [ab] | 5.48 [abc] | 0.39 [bc] |
| $K_{40} + K_{straw}$ | 233 [bc] | 94 [ab] | 91.0 [ab] | 25.4 [b] | 3.08 [b] | 5.20 [abc] | 0.37 [abc] |
| $K_{40} + K_{spray}$ | 313 [a] | 99 [ab] | 94.2 [a] | 27.0 [a] | 3.84 [a] | 6.22 [ab] | 0.38 [abc] |
| | | | Wet Season | | | | |
| $K_0$ | 400 [a] | 98 [a] | 84.1 [ab] | 25.8 [a] | 4.21 [c] | 4.35 [c] | 0.49 [a] |
| $K_{40}$ | 406 [a] | 103 [a] | 82.0 [b] | 25.0 [a] | 4.34 [bc] | 4.84 [abc] | 0.47 [a] |
| $K_{60}$ | 440 [a] | 99 [a] | 87.5 [a] | 25.9 [a] | 4.72 [bc] | 6.35 [ab] | 0.43 [a] |
| $K_{80}$ | 436 [a] | 110 [a] | 85.5 [ab] | 26.4 [a] | 5.27 [a] | 6.69 [a] | 0.43 [a] |
| $K_{straw}$ | 375 [a] | 101 [a] | 84.6 [ab] | 26.3 [a] | 4.55 [bc] | 4.99 [b] | 0.48 [a] |
| $K_{20} + K_{straw}$ | 408 [a] | 99 [a] | 86.8 [a] | 25.2 [a] | 4.39 [bc] | 4.26 [c] | 0.51 [a] |
| $K_{30} + K_{straw}$ | 440 [a] | 104 [a] | 86.6 [ab] | 26.8 [a] | 5.17 [ab] | 5.89 [abc] | 0.47 [a] |
| $K_{40} + K_{straw}$ | 460 [a] | 104 [a] | 86.9 [a] | 27.0 [a] | 4.94 [b] | 5.37 [abc] | 0.48 [a] |
| $K_{40} + K_{spray}$ | 444 [a] | 102 [a] | 85.1 [ab] | 26.9 [a] | 5.35 [a] | 5.88 [ab] | 0.48 [a] |

Values denoted with the same letter are not significantly different at $p < 0.05$ using Duncan's Multiple Range Test.

*3.3. Grain Yields*

The grain and straw yield results showed significant differences among the treatments in both seasons (Table 3). The yields in the WS were higher than in the DS. In the DS, the highest grain yields were recorded in $K_{40} + K_{spray}$ (3.84 t ha$^{-1}$), with 43% and 36% higher yields than the control ($K_0$) and $K_{straw}$, respectively. However, the grain yield in $K_{40} + K_{spray}$ was on par with treatments $K_{80}$, i.e., the highest dose of inorganic K fertilizer (80 kg $K_2O$ ha$^{-1}$) alone, and $K_{30} + K_{straw}$, i.e., integrated use of inorganic K fertilizer (30 kg $K_2O$ ha$^{-1}$) and straw (45 kg $K_2O$ ha$^{-1}$). Similar results were recorded in W.S.; the highest yields were produced in $K_{40} + K_{spray}$ (5.35 t ha$^{-1}$), significantly higher than $K_0$, $K_{20} + K_{straw}$, $K_{40}$, and $K_{60}$. However, it was on par with the treatments with the highest application of fertilizer dose, i.e., $K_{80}$ and $K_{30} + K_{straw}$. The lowest yield was recorded in $K_0$ in both seasons and was lower by more than 1 t ha$^{-1}$ than $K_{40} + K_{spray}$.

A similar trend was observed with straw yields in both seasons (Table 3). The straw yields were statistically higher in $K_{40} + K_{spray}$ and $K_{80}$ than $K_{40}$ and $K_0$ in DS and $K_{20} + K_{straw}$ and $K_0$ in WS. However, the other remaining treatments produced a similar straw yield during both seasons. The harvest indices were higher in WS than in DS, and there was no significant difference among the treatments in WS (Table 3).

The results also reported that the K response in yield increases was higher in $K_{40} + K_{spray}$, 42.7% in DS, and 27.1% in WS over the control ($K_0$) treatment. The second-

highest response was obtained with 60 kg $K_2O$ $ha^{-1}$ as fertilizer in DS (30.5%) and 30 kg $K_2O$ $ha^{-1}$ as fertilizer and straw (22.8%) in WS. The third-highest response was obtained in the treatment that received 30 kg $K_2O$ $ha^{-1}$ through fertilizer and straw in DS (29.4%) and 40 kg $K_2O$ $ha^{-1}$ through fertilizer and straw in WS (17.3%). The lowest response was obtained in treatment with straw only in DS and 40 Kg $K_2O$ $ha^{-1}$ in WS (Table 3).

### 3.4. Plant K Uptake as Affected by K Management at M.T., P.I., and Harvest Stages

The K uptake in DS was less than WS in all the stages. The maximum K uptake occurred between the PI and harvest stage in D.S., whereas it was more within the M.T. to P.I. stage in WS. K uptake varied significantly within treatments at all the growth stages (Table 4). During DS at the M.T. stage, K uptake varied from 13.4 kg $ha^{-1}$ ($K_{straw}$) to 19.1 kg $ha^{-1}$ ($K_{40} + K_{spray}$). $K_{40} + K_{spray}$ followed by $K_{60}$ accumulated a significantly higher K than $K_{20} + K_{straw}$, $K_{30} + K_{straw}$, and $K_{straw}$. At P.I., a similar trend was observed where $K_{40} + K_{spray}$ and $K_{60}$ had a similar K uptake and were significantly higher than $K_0$. However, as anticipated, straw had more K uptake at harvest than grain, and a consistent trend was observed in which $K_{80}$, followed by $K_{40} + K_{spray}$, had significantly higher values of K uptake than that of $K_{40}$, $K_{20} + K_{straw}$, and $K_0$. A similar trend was recorded with total K uptake. However, grain $K_{40} + K_{spray}$ had a similar K uptake with treatments $K_{80}$ and $K_{30} + K_{straw}$.

**Table 4.** Effect of potassium management practices on potassium uptake (kg $ha^{-1}$) by plants at different crop growth stages in dry and wet seasons.

| Treatment | Mid Tillering | Panicle Initiation | Harvest | | |
|---|---|---|---|---|---|
| | | | Straw | Grain | Total |
| *Dry Season* | | | | | |
| $K_0$ | 16.12 [abc] | 43.37 [c] | 65.68 [d] | 7.29 [c] | 74 [d] |
| $K_{40}$ | 17.40 [ab] | 49.73 [bc] | 90.52 [cd] | 10.59 [bc] | 100 [cd] |
| $K_{60}$ | 18.46 [a] | 67.53 [a] | 104.62 [abc] | 12.29 [b] | 117 [abc] |
| $K_{80}$ | 17.86 [ab] | 52.68 [bc] | 133.75 [a] | 13.34 [ab] | 147 [a] |
| $K_{straw}$ | 13.38 [c] | 46.15 [bc] | 99.03 [bc] | 12.13 [b] | 112 [bc] |
| $K_{20} + K_{straw}$ | 14.35 [b] | 49.21 [bc] | 78.35 [cd] | 13.31 [b] | 92 [cd] |
| $K_{30} + K_{straw}$ | 14.79 [b] | 46.11 [bc] | 108.87 [abc] | 14.62 [ab] | 123 [abc] |
| $K_{40} + K_{straw}$ | 16.57 [ab] | 53.02 [bc] | 97.28 [bcd] | 10.47 [bc] | 107 [c] |
| $K_{40} + K_{spray}$ | 19.08 [a] | 56.74 [ab] | 121.25 [ab] | 17.66 [a] | 139 [ab] |
| *Wet Season* | | | | | |
| $K_0$ | 24.58 [a] | 67.93 [ab] | 73.56 [c] | 13.23 [c] | 87 [d] |
| $K_{40}$ | 24.66 [a] | 62.04 [b] | 83.47 [bc] | 13.49 [c] | 97 [cd] |
| $K_{60}$ | 26.41 [a] | 83.51 [ab] | 130.49 [a] | 17.42 [ab] | 148 [a] |
| $K_{80}$ | 30.65 [a] | 80.42 [ab] | 138.67 [a] | 16.89 [abc] | 156 [a] |
| $K_{straw}$ | 21.93 [a] | 77.35 [ab] | 88.62 [bc] | 14.62 [bc] | 103 [bcd] |
| $K_{20} + K_{straw}$ | 29.18 [a] | 66.30 [ab] | 72.90 [c] | 14.24 [bc] | 87 [d] |
| $K_{30} + K_{straw}$ | 22.88 [a] | 82.17 [ab] | 120.93 [ab] | 16.79 [abc] | 138 [abc] |
| $K_{40} + K_{straw}$ | 22.84 [a] | 79.52 [ab] | 99.46 [abc] | 18.72 [a] | 118 [abcd] |
| $K_{40} + K_{spray}$ | 23.38 [a] | 85.67 [a] | 124.18 [ab] | 18 [ab] | 142 [ab] |

Values denoted with the same letter are not significantly different at *p* < 0.05 using Duncan's Multiple Range Test.

During WS at the P.I. stage, $K_{80}$, $K_{60}$, and $K_{40} + K_{spray}$ had significantly higher K uptake over $K_{20} + K_{straw}$ and $K_0$. At harvest, $K_{60}$, $K_{80}$, and $K_{40} + K_{spray}$ had significantly higher straw and total K uptake than $K_{40}$ and $K_0$. However, in grain, the highest K uptake was found in $K_{40} + K_{straw}$ followed by $K_{40} + K_{spray}$, and these had significantly higher K uptakes than $K_{40}$ and $K_0$ (Table 4).

### 3.5. Nitrogen and Phosphorus Uptake

The N and P uptake was higher in WS than in DS (Table 5). In DS, the N uptake was highest (75.52 kg ha$^{-1}$) in $K_{40} + K_{spray}$ and lowest (50.03 kg ha$^{-1}$) in $K_{straw}$. A similar trend was found in the WS; the N uptake was highest (94.39 kg ha$^{-1}$) in $K_{40} + K_{spray}$ and lowest (67.50 kg ha$^{-1}$) in the control. During both seasons, all treatments varied significantly. In the case of P uptake, during DS, $K_{40} + K_{spray}$ had the highest uptake and was significantly higher by 78%, 35%, and 52% than $K_0$, $K_{40}$, and $K_{straw}$, respectively. During WS, $K_{40} + K_{spray}$ performance was on par with $K_{60}$, $K_{80}$, and $K_{30} + K_{straw}$ and significantly superior over $K_0$, $K_{40}$, $K_{straw}$, and $K_{20} + K_{straw}$.

**Table 5.** Effect of potassium management practices on nitrogen and phosphorus uptake (kg ha$^{-1}$) at the harvest stage in dry and wet seasons.

| Treatment | Dry Season | | Wet Season | |
|---|---|---|---|---|
| | Nitrogen | Phosphorus | Nitrogen | Phosphorus |
| | (Nutrient Uptake) kg ha$^{-1}$ | | | |
| $K_0$ | 50.0 [c] | 10.1 [d] | 67.5 [c] | 21.5 [d] |
| $K_{40}$ | 57.0 [bc] | 13.3 [bcd] | 70.2 [c] | 23.8 [cd] |
| $K_{60}$ | 63.2 [abc] | 14.0 [bc] | 82.6 [abc] | 32.9 [a] |
| $K_{80}$ | 68.7 [ab] | 17.7 [a] | 90.8 [ab] | 32.9 [a] |
| $K_{straw}$ | 54.2 [bc] | 11.8 [cd] | 78.6 [bc] | 25.7 [bcd] |
| $K_{20} + K_{straw}$ | 57.2 [bc] | 14.9 [abc] | 71.8 [c] | 21.5 [d] |
| $K_{30} + K_{straw}$ | 64.1 [bac] | 16.1 [ab] | 87.8 [ab] | 31.0 [ab] |
| $K_{40} + K_{straw}$ | 57.0 [bc] | 13.8 [bc] | 80.6 [abc] | 28.3 [abcd] |
| $K_{40} + K_{spray}$ | 75.5 [a] | 17.9 [a] | 94.4 [a] | 30.6 [abc] |

Values denoted with the same letter are not significantly different at $p < 0.05$ using Duncan's Multiple Range Test.

### 3.6. K/N and K/P Ratios

An attempt was made to use the nutrient ratio concept to identify the sufficiency or deficiency of nutrients in plants. Both the K/N as well as K/P ratio values were higher in DS as compared to WS. The K/N ratio of the plant sample in DS did not vary significantly, while the K/P ratio had significant differences. $K_{straw}$ had the highest K/P ratio in DS. In contrast to DS during WS, the K/N ratio varied considerably. The maximum K/N ratio was recorded with $K_{60}$, which was statistically superior over $K_0$ and $K_{straw}$ only (Table 6).

**Table 6.** Effect of different K management practices on K/N, K/P in dry and wet seasons.

| Treatment | Dry Season | | Wet Season | |
|---|---|---|---|---|
| | K/N | K/P | K/N | K/P |
| $K_0$ | 1.47 [a] | 7.3 [bc] | 1.29 [b] | 4.04 [a] |
| $K_{40}$ | 1.76 [a] | 7.54 [bc] | 1.38 [ba] | 4.08 [a] |
| $K_{60}$ | 1.85 [a] | 8.33 [ab] | 1.79 [a] | 4.5 [a] |
| $K_{80}$ | 2.14 [a] | 8.34 [ab] | 1.71 [ab] | 4.72 [a] |
| $K_{Straw}$ | 2.07 [a] | 9.47 [a] | 1.31 [b] | 4.02 [a] |
| $K_{20} + K_{straw}$ | 1.61 [a] | 6.18 [c] | 1.21 [b] | 4.06 [a] |
| $K_{30} + K_{straw}$ | 1.92 [a] | 7.66 [abc] | 1.57 [ab] | 4.44 [a] |
| $K_{40} + K_{straw}$ | 1.88 [a] | 7.79 [abc] | 1.47 [ab] | 4.18 [a] |
| $K_{40} + K_{spray}$ | 1.84 [a] | 7.76 [abc] | 1.51 [ab] | 4.65 [a] |

Values denoted with same letter are not significantly different at $p < 0.05$ using Duncan's Multiple Range Test.

### 3.7. Potassium Harvest Index (KHI), Agronomic Efficiency (A.E.), and Recovery Efficiency (RE)

The KHI did not vary statistically during the study seasons (Table 7). Applied K's AE was highest with $K_{40} + K_{spray}$ treatment during both seasons. However, during DS, the AE of $K_{40} + K_{spray}$ was only significantly higher than $K_{straw}$ in contrast to W.S., where the AE was considerably higher than all other treatments. A similar trend was

recorded with recovery efficiency; during DS, $K_{40} + K_{spray}$ was markedly higher than $K_{40}$, $K_{20} + K_{straw}$, $K_{30} + K_{straw}$, and $K_{40} + K_{straw}$. However, in WS, $K_{40} + K_{spray}$ was only significantly different from the $K_{20} + K_{straw}$ treatment (Table 7).

**Table 7.** Effect of different potassium management practices on potassium harvest index (KHI), agronomic efficiency (AE) and recovery efficiency (RE) in dry and wet seasons.

| Treatment | Dry Season | | | Wet Season | | |
|---|---|---|---|---|---|---|
| | KHI | AE | RE | KHI | AE | RE |
| $K_0$ | 0.10 [a] | | | 0.15 [a] | | |
| $K_{40}$ | 0.11 [a] | 12.83 [ab] | 66.6 [b] | 0.16 [a] | 3.33 [b] | 25.4 [ab] |
| $K_{60}$ | 0.11 [a] | 13.56 [ab] | 71.8 [ab] | 0.12 [a] | 8.56 [b] | 101.9 [ab] |
| $K_{80}$ | 0.09 [a] | 9.13 [ab] | 92.2 [ab] | 0.11 [a] | 8.21 [b] | 86.0 [ab] |
| $K_{straw}$ | 0.11 [a] | 2.78 [b] | 85.6 [ab] | 0.15 [a] | 7.63 [b] | 36.5 [ab] |
| $K_{20} + K_{straw}$ | 0.15 [a] | 7.26 [ab] | 28.8 [b] | 0.17 [a] | 2.77 [b] | 0.50 [b] |
| $K_{30} + K_{straw}$ | 0.12 [a] | 10.51 [ab] | 66.3 [b] | 0.12 [a] | 12.80 [b] | 67.9 [ab] |
| $K_{40} + K_{straw}$ | 0.10 [a] | 4.47 [ab] | 39.4 [b] | 0.16 [a] | 8.55 [b] | 36.9 [ab] |
| $K_{40} + K_{spray}$ | 0.13 [a] | 27.56 [a] | 157.5 [a] | 0.13 [a] | 27.40 [a] | 133.1 [a] |

Values denoted with same letter are not significantly different at $p < 0.05$ using Duncan's Multiple Range Test.

### 3.8. Soil Available K at M.T., P.I., and Harvest Stages

The available soil K in the DS and WS varied significantly at different growth stages of the crop (Table 8). For DS, at the M.T. stage, the highest value was recorded in the treatment $K_{80}$, and the lowest value was in the treatment $K_0$. A similar trend was also found at the P.I. stage and maturity (Table 8). K applied through straw alone and combined with inorganic K fertilizer has a lower value of available K compared to the highest dose of inorganic K fertilizer. For WS, no particular trend was observed at different crop growth stages. At maturity, the highest value was recorded in treatment $K_{60}$, and the lowest value was in treatment $K_0$ (Table 8).

### 3.9. Soil Potassium Balance

Figure 2 represents the soil K balance for DS and WS in the rice–rice cropping system. There was a drastic difference in soil K balance in different treatments during seasons and years. The highest negative balance ($-135$ kg ha$^{-1}$) was recorded with $K_{40} + K_{spray}$ treatment, slightly higher than $K_0$ ($-113.9$ kg ha$^{-1}$). Similarly, $K_{40}$, $K_{60}$, and $K_{80}$ treatments had negative K balances. The highest positive K balance was recorded with the treatment $K_{40} + K_{straw}$, followed by $K_{20} + K_{straw}$ and $K_{30} + K_{straw}$, while applying straw alone ($K_{straw}$) maintained only a 7 kg ha$^{-1}$ K positive balance. All the treatments where straw was incorporated had a positive K balance.

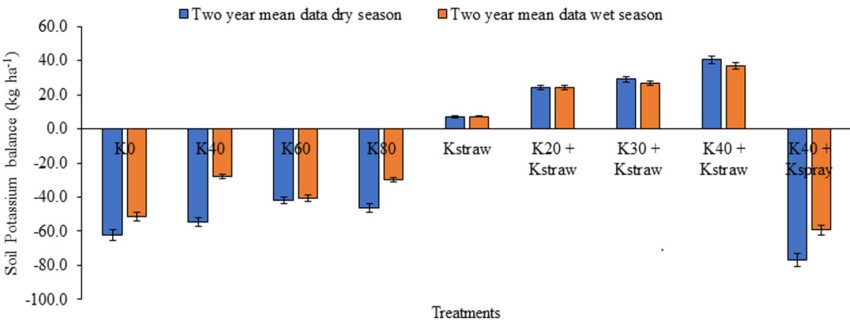

**Figure 2.** Potassium balance for dry and wet seasons in the rice-rice cropping system.

**Table 8.** Effect of different potassium management practices on available soil potassium (kg ha$^{-1}$) at different growth stages of rice crop during dry and wet seasons.

| Treatment | Maximum Tillering | | Panicle Initiation | | Harvest | |
|---|---|---|---|---|---|---|
| | 0–15 cm | 15–30 cm | 0–15 cm | 15–30 cm | 0–15 cm | 15–30 cm |
| | Dry Season | | | | | |
| $K_0$ | 68.4 [d] | 42.1 [b] | 56.2 [d] | 45.9 [c] | 47.9 [c] | 46.8 [b] |
| $K_{40}$ | 89.2 [c] | 46.9 [b] | 65.7 [c] | 56.1 [bc] | 51.1 [c] | 50.2 [ab] |
| $K_{60}$ | 104.0 [ab] | 67.4 [a] | 82.4 [ab] | 63.5 [b] | 74.6 [ab] | 52.5 [ab] |
| $K_{80}$ | 117.7 [a] | 78.4 [a] | 94.7 [a] | 72.2 [a] | 79.6 [a] | 62.5 [a] |
| $K_{Straw}$ | 108.5 [ab] | 57.1 [ab] | 87.5 [ab] | 52.8 [bc] | 58.5 [bc] | 48.1 [b] |
| $K_{20} + K_{Straw}$ | 84.3 [bc] | 42.2 [b] | 67.0 [bc] | 46.1 [c] | 50.9 [c] | 48.4 [b] |
| $K_{30} + K_{Straw}$ | 94.2 [bc] | 50.3 [ab] | 81.1 [ab] | 50.4 [bc] | 66.6 [b] | 47.0 [b] |
| $K_{40} + K_{Straw}$ | 90.3 [bc] | 64.6 [a] | 83.5 [ab] | 58.2 [bc] | 78.8 [a] | 47.3 [b] |
| $K_{40} + K_{Spray}$ | 92.0 [bc] | 48.7 [b] | 69.1 [bc] | 52.0 [bc] | 57.8 [bc] | 48.2 [b] |
| | Wet Season | | | | | |
| $K_0$ | 42.3 [d] | 33.1 [b] | 29.0 [d] | 28.0 [c] | 35.6 [c] | 46.9 [b] |
| $K_{40}$ | 52.3 [cd] | 39.9 [b] | 39.2 [c] | 33.9 [bc] | 47.6 [b] | 47.3 [b] |
| $K_{60}$ | 59.1 [c] | 38.2 [b] | 47.4 [b] | 43.7 [a] | 63.3 [a] | 70.9 [a] |
| $K_{80}$ | 71.6 [b] | 45.8 [ab] | 44.2 [bc] | 42.3 [a] | 46.0 [b] | 62.5 [a] |
| $K_{Straw}$ | 48.6 [cd] | 42.0 [b] | 55.8 [a] | 37.1 [b] | 45.2 [b] | 51.7 [b] |
| $K_{20} + K_{Straw}$ | 64.9 [bc] | 43.6 [b] | 37.0 [c] | 34.6 [b] | 48.1 [b] | 47.1 [b] |
| $K_{30} + K_{Straw}$ | 81.6 [a] | 51.3 [a] | 55.8 [a] | 46.1 [a] | 59.5 [a] | 59.4 [a] |
| $K_{40} + K_{Straw}$ | 66.9 [bc] | 55.3 [a] | 46.2 [b] | 43.2 [a] | 46.3 [b] | 54.0 [ab] |
| $K_{40} + K_{Spray}$ | 50.1 [cd] | 41.8 [b] | 39.0 [c] | 36.5 [b] | 43.7 [bc] | 50.5 [ab] |

Values denoted with the same letter are not significantly different at $p < 0.05$ using Duncan's Multiple Range Test).

### 3.10. GHG Estimation under Different Potassium Management Options

Indian agriculture has the potential to mitigate 85.5 Mt $CO_2$ eq per year, 80% of which would be delivered by cost-effective options, particularly fertilizer and crop residue management. Total GHGs emissions increased with the increased integration of residues with K application as compared to only incremental application of K; however, integration of K (40 kgha$^{-1}$) with $K_{spray}$ reduces the total GHGs emission by 3% as compared to $K_{40} + K_{straw}$ (Figure 3).

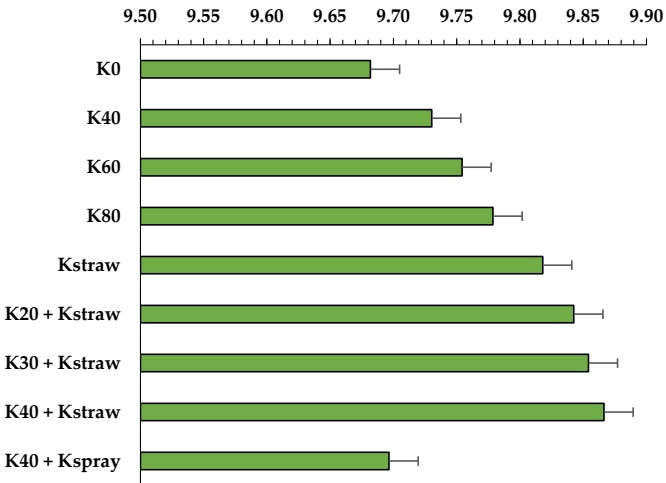

**Figure 3.** Effect of potassium management options on total GHGs emissions across the seasons.

## 4. Discussion

### 4.1. Effect of Potassium Management Strategy on Crop Yields

During the green revolution, around the 1970s, excessive use of inorganic fertilizers improved crop production to a large extent and ensured food security for the growing population. In recent decades, crop production has either stagnated or declined. This is due to a decline in soil quality and the imbalanced application of fertilizers [42]. Among all fertilizers, K is one of the important essential macronutrients which significantly impact crop growth and development. It contributes to many regulatory processes in rice such as improving grain quality by translocating photosynthetic products and other plant metabolites [43,44]. Farmers use inorganic K fertilizers such as the muriate of potash (MOP) and the sulphate of potash (SOP) to meet their crop demands. These K fertilizers are costly, unavailable at the time of requirement, and also have ill effects on soil health. So, alternative options for K management have been discovered by researchers and evaluated in several soils and crops. It is reported that integrated application of inorganic K fertilizers in soil along with organic sources such as rice straw or foliar spray can be adopted for K management options in rice [45,46]. As shown in the study, $K_{40} + K_{spray}$ had a better impact on rice yields under both seasons, i.e., DS and WS in acid soils of eastern India compared to treatments with no fertilization or straw alone or with the recommended dose of fertilizer (40 kg K ha$^{-1}$). The inorganic K applied with straw did not perform well in D.S., which might be due to the slow decomposition of straw in prevailing low temperatures and low rainfall as well as humidity during the crop period. However, in W.S., there was a significant improvement in grain yields in $K_{30} + K_{straw}$ and $K_{40} + K_{straw}$, which might be due to the faster decomposition of straw in the prevailing higher temperatures and humidity, along with the high rainfall during the crop period. Similar results were reported by [47], where straw addition in soil started to improve grain yields only after the third season.

Our study showed that the yields are lower in DS than in WS. This can largely be attributed to poor tillering and a reduced number of spikelets per panicle, as shown in the results. The inadequacy of N and P due to less P diffusion and poor uptake and utilization of P in the early part of the growth of DS crops and the low temperature prevailing during the planting and tillering stage has resulted in lower yields. Graham et al. [48] reported that the applied N might have been lost through the leaching of irrigation water which is introduced more frequently after the planting of DS crops. Between DS and WS, leaching is at a minimum in WS because of the land characteristics. In contrast, it is greater in DS because, with a fall in the water table, there is a requirement of frequent irrigation, through which applied N might have lost from the root zone through leaching. To substantiate the above cause, an attempt was made to use the nutrient ratio concept to identify the sufficiency or deficiency of nutrients in plants. From the N and P absorption results compared to K, it may be inferred that there is less absorption of N and P. The ratio value is more in DS than in WS both in terms of content and total uptake, which may be a limiting factor for yield in DS. The biomass accumulation was also more in WS than in DS as a result of slow plant growth due to the low temperature during the initial growth period of the DS crop.

### 4.2. Effect of Potassium Management Strategy on Potassium Uptake

Son et al. [49] found that K content and uptake in plants increase with the growth of the stages of the crop. K content of WS in the M.T. and P.I. stages was comparatively higher than DS, while it was similar at the maturity stage of both seasons. However, K content in the straw was higher in the treatments which received a higher K fertilizer dose than the lower-K-dose or straw-incorporated treatments. In DS, grains accumulated 10.47 to 34.10 kg K ha$^{-1}$ compared to 65.68–133.75 kg K ha$^{-1}$ in straw. Thus, 9.06–14.42% of the total K is partitioned into the grain. Whereas in WS, at maturity, grains accumulated 10.86 to 16.34% of the total K. Hence, straw retained a significant portion of K absorbed by the plant. Even comparing the partitioning of K absorption to different periods of growth, it

was observed that in DS the treatments that yielded more absorbed 12–13% of total K within the M.T. stage, 23–27% during the M.T. to P.I. stage, and 59–64% within PI to maturity as compared to 16–18% of total K within MT, 38–44% during M.T. to P.I., and 40–44% within P.I. and maturity in WS. In a study, [50] reported 24% of the total K absorbed within 37 days, and 41% during 38–58 days which matched with the results of WS. From the partitioning data, it is clear that in the early vegetative growth period during DS, the crop absorbed less K than that in WS. This suggests that for producing higher yields, around 60% K needs to be absorbed after PI in DS and around 40% in each of the two stages M.T. to P.I. and P.I. to maturity in WS. Therefore, poor absorption of N, P, and K might be the reason for the reduced number of panicles in DS, as K absorbed up to the M.T. stage is used to increase the number of panicles and the number of grains per panicle [51,52]. Further, in DS, the number of effective tillers per square meter was also low.

### 4.3. Effect of Potassium Management Strategy on Potassium Efficiency and Balance

Among K efficiencies, the AE and RE of applied K were significantly greater in $K_{40} + K_{spray}$ (basal dose of K along with foliar spray); this might be due to better absorption and consequent assimilation of nutrients supplied through foliar application at the P.I. stage. This improved tillering and grain yields of rice; however, even with double the recommended fertilizer dose, $K_{80}$ could not be performed. Similarly, K application through the straw and graded dose of fertilizer could not meet the K requirement; subsequently, there was a poor K/N and K/P ratio. Batra et al. [53] also reported that the foliar spray of K affects the N uptake by rice and increases the grain yields.

There was an eye-catching difference in K balance based on the treatments. Although $K_{40} + K_{spray}$ was more effective in improving grain yields than other treatments, it adversely affected soil sustainability by removing 135 kg K ha$^{-1}$ from the soil. Even double the recommended fertilizer dose had adversely affected soil sustainability by removing 76.2 kg K ha$^{-1}$ from the soil, whereas adding straw improves the K content in the soil (Figure 2). This explains that the straw residue addition positively impacts soil sustainability. Hence, applying organics in the field provides enhanced soil quality/health. Adding straw residues alone or combined with inorganic fertilizer has a positive K balance [54]. This is due to the improved nutrient-holding capacity when soil is added with organics. Application of inorganic K fertilizer alone resulted in a negative soil K balance. A dynamic equilibrium exists between exchangeable soil K and non-exchangeable K. A portion of non-exchangeable K can be transformed into exchangeable K when exchangeable K is lower than a threshold concentration [55–57]. The significant decrease in the inorganic K treatments may be due to the transformation of slowly available K into available K, which had been absorbed and removed by crops. In addition, the soil weathering induced by crop roots or microorganisms also could contribute considerable amounts of K to the soil solution.

### 4.4. Effect of Potassium Management Strategy on Greenhouse Gases

In this study, $CH_4$ emission from rice production considers pre-season water status, current water regimes, soil organic C, organic amendment, and their interaction with various soil and climatic factors [58]. This approach allows for a high level of sensitivity to climatic conditions and soil properties, especially to soil pH, and hence a better representation of growing conditions in India [59]. The emissions associated with the production and transportation of fertilizer were included in our study. $CH_4$ emissions from rice–rice cropping systems are highly dependent on the amount of residue recycled under continuously flooded conditions. Potassium application can promote the development of rice aerenchyma and enhance the gas transfer from the bottom soil to the atmosphere, thus raising $CH_4$ and $N_2O$ emissions and thus total GHGs emissions. Furthermore, the addition of crop residue with incremental doses of K aggravate the GHGs emissions, as reflected in our study.

## 5. Conclusions

It is essential to provide rice crops with sufficient available K at an appropriate time and as per the demands for better crop yields and soil sustainability. This study found alternative K management options that can reduce the dose of chemical K fertilizer when applied along with straw as a supplemental source of K and foliar application of K in the rice–rice cropping system. This study also concluded that the treatments involving the incorporation of straw alone or integrated with soil-applied fertilizer K had positive soil K balance. Though the treatment with foliar spray of K fertilizer yielded the maximum and performed well on many parameters, it also had the maximum negative soil K balance. In terms of yield and soil sustainability, integrated use of inorganic K fertilizer (30 kg $K_2O$ ha$^{-1}$) and straw (45 kg $K_2O$ ha$^{-1}$) can be a better K management option for puddled, transplanted rice grown in acidic soils of eastern coastal India. Incremental doses of K along with straw aggravate the GHGs emission in rice–rice cropping systems.

**Author Contributions:** Conceptualization, K.K.R.; data curation, S.M., K.K.R. and A.M.; funding acquisition, S.Y. and S.S.; investigation, S.M. and A.K.M.; methodology, K.K.R.; project administration, C.K. and S.Y.; resources, C.K. and S.S.; supervision, K.K.R. and S.S.; validation, K.K.R. and S.S.; writing—original draft, S.M. and K.K.R.; writing—review and editing, R.P., and A.M. All authors have read and agreed to the published version of the manuscript.

**Funding:** The research was funded by IRRI, New Delhi.

**Institutional Review Board Statement:** Not applicable.

**Informed Consent Statement:** Not applicable.

**Data Availability Statement:** As and when requested by the readers.

**Acknowledgments:** We sincerely acknowledge the fellowship awarded by the Cereal Systems Initiative for South Asia (CSISA) and the OUAT collaborative research project of IRRI for carrying out the research work.

**Conflicts of Interest:** The authors declare no conflict of interest.

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
