# Peer review of "Evaluation of Different Potassium Management Options under Prevailing Dry and Wet Seasons in Puddled, Transplanted Rice"

_sustainability, doi:10.3390/su15075819_

Round 1

Reviewer 1 Report

Well-presented and supported/discussed in light of earlier studies.

Authors need to mention:

1. year of study in abstract

2.How much straw was added to obtain 45 kg k2o

3. Better write objectives in running

Studies is good and yielding good recommendation and may be published as such. 

But in future authored need to study some more treatments such as only potassium nitrate, and potassium nitrate in combination with graded inorganic K level and also straw doses

Author Response

Here is a point-by-point response to the comments and concerns. 

  1. Authors need to mention the year of study in abstract.

Response: Thank you for pointing this. We have mentioned it in the abstract.

  1. How much straw was added to obtain 45 kg k2o

Response: We have added 3 t ha-1 straw. Generally, rice straw contains 1.5% of K as reported by many authors. We have added 3 tons of straw per hectare which is equivalent to 45 kg K2O per ha. Further, the recommended dose of K is 40 kg per ha, so, 3 t ha-1 straw was added which was almost equivalent to the recommended dose. We have mentioned this under the sub heading crop and nutrient management (2.4).

  1. Better write objectives in running

Response: Complied as per the suggestion

Studies is good and yielding good recommendation and may be published as such. 

But in future authored need to study some more treatments such as only potassium nitrate, and potassium nitrate in combination with graded inorganic K level and also straw doses

Response: We absolutely agree with you in this regard and looking forward to add more treatments to strengthen the findings in the future studies.

Reviewer 2 Report

Comments and Suggestions for Authors

The manuscript entitled “Evaluation of different potassium management options under prevailing dry and wet seasons in puddled transplanted rice” by S. Mohapatra et al.. I appreciate the great effort the author has put into this work, but some details need to be checked at present. The authors will have to make revisions if they want to publish the article. A minor revision is required for the reasons listed below:

The length of the article is too long, please carefully consider, delete wordy sentences.

Abstract:

1.     Abstract is too many words, usually no more than 250 words.

Materials and methods

2.     Please check formula (1). There is no need to mark units in the calculation formula. Please check formula (5) (6) “:”.

3.     Please explain in detail how to calculate CCAFS? Are the results reliable? How to calculate soil greenhouse gas emissions?

4.     Please explain why the yield is higher in the wet season than in the dry season?

5.     Please supplement the potassium content data in Kstraw and Kspray.

Results

6.     Table 5 seems to have been made improperly.

7.     3.10. There should be no literature citations.

Discussion

8.     Subheadings are recommended because there are too many words in the discussion.

Conclusion

9.     Finally, There is too much text in the conclusion, please simplify. The first sentences are more suitable for the introduction.

10.  The article has formatting problems, please check.

Author Response

Here is a point-by-point response to the comments and concerns. 

The manuscript entitled “Evaluation of different potassium management options under prevailing dry and wet seasons in puddled transplanted rice” by S. Mohapatra et al.. I appreciate the great effort the author has put into this work, but some details need to be checked at present. The authors will have to make revisions if they want to publish the article. A minor revision is required for the reasons listed below: 

The length of the article is too long, please carefully consider, delete wordy sentences.

Abstract:

  1. Abstract is too many words, usually no more than 250 words.

Response: Abstract is now concise to reduce wording

Materials and methods

  1. Please check formula (1). There is no need to mark units in the calculation formula. Please check formula (5) (6) “:”.

Response: The formula in equations 5 and 6 are ratios and unitless

  1. Please explain in detail how to calculate CCAFS? Are the results reliable? How to calculate soil greenhouse gas emissions?

Response: The Climate Change, Agriculture and Food Security (CCAFS) research programme of the CGIAR supports the development of user-friendly science based decision-making tools that support policy advisers to design policies that maximise GHG emission mitigation in agriculture. This tool has been named CCAFS-MOT (i.e. CCAFS-Mitigation Option Tool).

The CCAFS-MOT estimates GHG emissions from several upland crops, rice and livestock systems in different geographic regions and it ranks the most effective mitigation options for these different crops according based on current management practices, climate and soil characteristics worldwide. The tool joins several empirical models derived from IPCC to estimate GHG emissions and consider mitigation practices that are compatible with food production (Feliciano et al., 2017).

  1. Please explain why the yield is higher in the wet season than in the dry season?

Response: Mentioned in first paragraph of Page no. 14

  1. Please supplement the potassium content data in Kstrawand Kspray.

Response: For Kstraw, an additional dose of 45 kg K through straw. Generally, rice straw contains 1.5% of K as reported by many authors. We have added 3 tons of straw per hectare which is equivalent to 45 kg K2O per ha. Further, the recommended dose of K is 40 kg per ha, so, 3 t ha-1 straw was added which was almost equivalent to the recommended dose. For Kspray. 1% KNO3 is added which supplies around 2.5 kg K.

Results

  1. Table 5 seems to have been made improperly.

Response: Thank you for pointing this. We have made necessary changes in the revised manuscript.

  1. 3.10. There should be no literature citations.

Response: We have rectified this in the revised manuscript.

Discussion

  1. Subheadings are recommended because there are too many words in the discussion.

Response: Complied as suggested.

Conclusion

  1. Finally, There is too much text in the conclusion, please simplify. The first sentences are more suitable for the introduction.

Response: Conclusion is now concise and simplified

  1. The article has formatting problems, please check.

Response: Thank you for pointing this. We have made necessary changes in the revised manuscript.

Reviewer 3 Report

1.Please follow the prescribed pattern of the journal through out the manuscript

2. Write full forms of first time used abbreviation

3.How seedling age is related to DS. DS means direct seeding of rice seed in the puddled field. However, in WS seedlings of certain age are transplanted. Please elaborate

4.The methodology is related to WS transplanted. What about DS?

5.Had rice straw been added during both DS & WS?. Please clarify

6.  As per journal’s pattern, results and discussion should be combined in one heading

 7.What was the initial K status of the experimental soil?

8.References are not formatted as per journal’s pattern

Author Response

Here is a point-by-point response to the comments and concerns. 

Please follow the prescribed pattern of the journal through out the manuscript

Response: Thank you for pointing this. We have made necessary changes in the revised manuscript.

  1. Write full forms of first time used abbreviation

Response: We have rectified this in the revised manuscript.

3.How seedling age is related to DS. DS means direct seeding of rice seed in the puddled field. However, in WS seedlings of certain age are transplanted. Please elaborate

The methodology is related to WS transplanted. What about DS?

Response: Actually, in our study DS symbolizes the Dry Season.

5.Had rice straw been added during both DS & WS? Please clarify

Response: Yes, rice straw been added during both DS and WS.

  1. As per journal’s pattern, results and discussion should be combined in one heading

Response: We have referred the recent format as instructed by the journal where it was mentioned that results and discussion may be combined. https://www.mdpi.com/journal/sustainability/instructions#preparation ( this link may please be referred)

 7.What was the initial K status of the experimental soil?

 Response: The initial available K is 57.95 kg ha-1. We have mentioned this under the sub heading field site (2.1).

8.References are not formatted as per journal’s pattern

Response: Thank you for pointing this. We have rectified this in the revised manuscript.

Round 2

Reviewer 2 Report

accepct

Reviewer 3 Report

The author has revised the manuscript as per suggestions/comments given in the manuscript.